# Spectrum and Incidence of Adverse Reactions Post Immunization in the Taiwanese Population (2014–2019): An Analysis Using the National Vaccine Injury Compensation Program

**DOI:** 10.3390/vaccines12101133

**Published:** 2024-10-03

**Authors:** Wan-Chun Lai, Chin-Hui Yang, Yhu-Chering Huang, Nan-Cheng Chiu, Chih-Jung Chen

**Affiliations:** 1Division of Pediatric Infectious Diseases, Departments of Pediatrics, Chang Gung Memorial Hospital, Taoyuan 333, Taiwan; debbie61640@gmail.com (W.-C.L.); ychuang@cgmh.org.tw (Y.-C.H.); 2Taiwan Centers for Disease Control, Ministry of Health and Welfare, Taipei 100, Taiwan; inf@cdc.gov.tw; 3School of Medicine, College of Medicine, Chang Gung University, Taoyuan 333, Taiwan; 4Department of Pediatrics, MacKay Children’s Hospital, Taipei 104, Taiwan; ncc88@mmh.org.tw; 5Molecular Infectious Diseases Research Center, Chang Gung Memorial Hospital, Taoyuan 333, Taiwan

**Keywords:** vaccine reactogenicity, adverse reactions, adverse events following immunization, vaccine injury compensation program

## Abstract

**Background**: Post-marketing surveillance is crucial for gathering data on vaccine reactogenicity, enhancing public trust in immunization, and promoting vaccine uptake. This study aims to characterize adverse events following immunization (AEFIs) and estimate the incidence rates of adverse reactions (ARs) associated with vaccines included in Taiwan’s Expanded Program on Immunization (EPI). This study utilizes data from Taiwan’s Vaccine Injury Compensation Program (VICP). **Methods**: Vaccine injury claims submitted to the VICP between 2014 and 2019 were analyzed. ARs were defined as AEFIs adjudicated as “related” or “indeterminate” by the VICP committee. Data on the annual number of vaccine doses administered were obtained from the Taiwan CDC, which helped calculate the AR incidence rates. **Results**: A total of 491 AEFI claims were reviewed, with 327 (66.6%) categorized as ARs. The AEFIs were mainly associated with the Bacillus Calmette–Guérin (BCG) vaccine (43.4%) and the seasonal influenza vaccine (22.0%). Most EPI vaccines demonstrated low AR incidence rates, ranging from 1.68 to 13.6 per million doses, with the exception of BCG, which exhibited 162.5 ARs per million doses. Shifting BCG immunization from below 5 months to at least 5 months reduced osteomyelitis incidence significantly, from 41.4 to 7.9 (*p* = 0.0014), but increased abscess and lymphadenitis cases. **Conclusions**: EPI vaccines in Taiwan are highly safe, with minimal AR incidences in the general population. The BCG vaccine remains an exception, occasionally causing severe ARs like osteomyelitis. Adjusting the immunization schedule to an older age may mitigate some of these adverse effects.

## 1. Introduction

The development and widespread use of effective vaccines play a fundamental role in controlling numerous serious and potentially lethal contagious diseases worldwide. The impact of vaccines is significant, with estimates suggesting that global immunization programs save four to five million lives annually [1]. Despite their high safety profiles, adverse events following immunization (AEFIs) are frequently encountered, although most are mild and resolved spontaneously within a short period. Serious adverse events (SAEs), while rare, can be associated with severe morbidity, long-term sequelae, or even death. Establishing a causal relationship between SAEs and vaccination on an individual case basis can be challenging, leading to legal conflicts between affected individuals, vaccine manufacturers, healthcare institutions, and governments. Moreover, safety concerns and misinformation about vaccines have emerged as critical issues contributing to vaccine hesitancy, suboptimal vaccine coverage, and the resurgence of vaccine-preventable diseases [2]. Data concerning the types and incidence of AEFIs, with their causality linked to vaccines (hereafter referred to as adverse reactions or ARs), are crucial for building public trust in vaccines and promoting vaccine uptake. However, such information is lacking in many countries.

In Taiwan, several systems are available for reporting AEFIs. These include the adverse drug reaction (ADR) reporting system operated by the Taiwan Drug Relief Foundation and the Taiwan National ADR Reporting Center, which is affiliated with the Taiwan Food and Drug Administration [3]. Additionally, the Vaccine Adverse Event Reporting System (VAERS) is managed by the Taiwan Centers for Disease Control (CDC), and there is also the National Vaccine Injury Compensation Program (VICP) [4]. All of these systems utilize passive reporting from individuals, healthcare providers, or pharmacists.

The VICP is considered a potential resource for the post-marketing surveillance of vaccine ARs. The Federal Republic of Germany established the first vaccine injury compensation system in 1961 [2]. In the United States, the surge in litigation related to vaccination in the 1970s led to vaccine shortages and a decline in the national vaccination rates as manufacturers withdrew production [5]. Concerns about a resurgence of vaccine-preventable diseases prompted the establishment of the national VICP between 1986 and 1988 as a public health safeguard in the United States [5,6]. In Taiwan, the VICP was introduced in 1988 following a serious adverse event where a boy developed polio-like symptoms after receiving the oral polio vaccine [7]. The program aims to raise awareness of vaccine safety, encourage immunization, and maintain a high vaccination coverage. It operates under a “no-fault” compensation scheme, compensating injured patients or their families through government funding once a causal link between an adverse event and immunization is established [8,9,10].

The Vaccine Adverse Event Reporting System (VAERS) is another channel for reporting AEFIs in Taiwan. However, the national VICP offers several advantages over the VAERS database for analyzing vaccine ARs in Taiwan. Firstly, the VICP was established in 1988, predating the VAERS. Secondly, due to public debates over high-profile AEFI cases adjudicated as “unrelated” by the VICP during the 2009 influenza A H1N1 pandemic, the VICP is more familiar to the Taiwanese public. Thirdly, while VAERS reporting is restricted to healthcare providers, the VICP is open to the public, allowing any Taiwanese citizen to file a claim of vaccine injury [11]. Such claims are encouraged by the Taiwanese government and can be easily submitted via an online application form (https://www.cdc.gov.tw/Vicp/Fill ; accessed on 1 October 2024). Finally, after registration, a series of processes ensue, including local health bureaus collecting adverse event-related medical records and an expert committee determining causality. The causal relationship between AEFIs and vaccination is not determined in the VAERS. Thus, the VICP represents the most comprehensive database for the post-marketing surveillance of vaccine ARs in Taiwan. In this study, we aim to analyze the incidence and types of ARs associated with various vaccines using the VICP database from 2014 to 2019, before the COVID-19 pandemic.

## 2. Materials and Methods

### 2.1. Ethical Approval

This study was approved by the Research and Ethics Committee of Chang Gung Memorial Hospital (IRB: 202002607B1).

### 2.2. Claims to the Vaccine Injury Compensation Program (VICP) 

The National Vaccine Injury Compensation Program (VICP) in Taiwan was established under Article 30 of the Communicable Disease Control Act. This program encompasses all officially approved vaccines used for disease prevention, irrespective of whether they are publicly funded or self-paid. In cases of suspected vaccine-related injury, claims may be filed within two years of becoming aware of the adverse event following immunization (AEFI) or within five years from the first appearance of any AEFI symptoms. Claims can be submitted by the injured individuals themselves, guardians of individuals under 20 years old, or heirs of deceased individuals.

The local health bureau manages the claim process and is responsible for gathering all pertinent medical records related to the adverse event. These include clinical manifestations, laboratory data, examination reports, medical history, disease progression, treatment outcomes, and details about the vaccines involved. At least two medical experts from the VICP working group independently review these records and the relevant literature to assess the causality of the AEFI. If there is disagreement between the two experts, a third expert is consulted. The final decision on causality is reached collectively during a review conference attended by all members of the VICP working group, which comprises clinical physicians, pharmacists, pathologists, legal professionals, and impartial community members [7,12].

AEFI causality is classified as “related”, “indeterminate”, or “unrelated” to vaccination. Financial compensation is provided for cases adjudicated as “related” or with “indeterminate” causality. Compensation is also available for unrelated cases involving deceased individuals who underwent autopsy, those who had medical procedures to determine causality, and pregnant women who experienced stillbirth or miscarriage [13]. On average, the resolution process from claim filing to adjudication takes approximately six months, with claimants having the right to appeal within 30 days if dissatisfied with the outcome.

Following Wang’s publication of updates to Taiwan’s VICP in 2014 [7], this study collected data from claims adjudicated between 2014 and 2019. The VICP database records include claimants’ age, sex, and residence, types of vaccines involved, dates of immunization, reported symptoms, clinical diagnoses, duration from vaccination to symptom onset, adjudication results, and compensation amounts. The data were collected from the VICP database. For this study, AEFIs adjudicated as “related” or “indeterminate” were classified as ARs to the vaccines. Adjudication outcomes are periodically published online by the Taiwan Centers for Disease Control [14]. Claims adjudicated from 2014 to 2019 were reviewed and analyzed in this study.

### 2.3. Definitions

Data on the total administered doses of each vaccine during the study period, as provided by the Taiwan CDC (Appendix A), were used to estimate the AR incidence for each vaccine. The immunization schedule in Taiwan is listed in Appendix A. If multiple vaccine types were administered simultaneously and the specific offending vaccine could not be precisely identified, the AEFI was attributed to all administered vaccines. The interval from vaccination to AEFI onset was calculated in days, with symptoms occurring on the day of vaccination counted as 0.5 days.

### 2.4. Statistics

Statistical analysis was performed using GraphPad Prism 9.0.0 and Microsoft Excel © 2022 Microsoft Corporation. Medians were compared using the Mann–Whitney U test, while categorical variables were compared using the Chi-square test and Fisher’s exact test. A *p*-value of < 0.05 was considered statistically significant. 

## 3. Results

From 2014 to 2019, the Vaccine Injury Compensation Program (VICP) of Taiwan received and adjudicated a total of 491 claims, with an annual range of 42 to 122 claims (mean 81.8 claims). Of these, 327 claims (66.6%) were adjudicated as either “related” (231 claims, 47.0%) or “indeterminate” (96 claims, 19.6%), while the remaining 164 claims (33.4%) were adjudicated as “unrelated” (Table 1). The total compensation disbursed for all claims amounted to approximately 30 million New Taiwan Dollars (~1 million USD) during the 2014–2019 period. The Bacillus Calmette–Guerin (BCG) vaccine was the most frequently identified offending vaccine, with 213 claims, followed by the influenza vaccine, with 108 claims (Figure 1a and Table 1). The incidence of adverse reactions (ARs) was generally low (Figure 1b,c), averaging 6.0 ARs per million vaccine doses, except for the BCG vaccine, which showed an incidence of 162.5 ARs per million doses.

### 3.1. BCG Vaccine

Detailed AR data for the BCG vaccine can be found in Table 2 and Figure 2a. Abscess and lymphadenitis were the two most common ARs, typically observed at the vaccination site (left arm) and left axillary region but, occasionally, also at more distant sites such as the chest wall and supra- or infra-clavicular areas. Osteomyelitis was also notable, with an incidence of 22.9 episodes per million vaccine doses, accounting for 26.7% (55 out of 206) of all BCG-related ARs. BCG osteomyelitis mainly developed between 6 and 24 months post immunization, with an average onset interval of 15.9 ± 6.0 months (Figure 2b, Table 2). Lower-limb long bones were the most affected, followed by the sternum/ribs and the foot, ankle, or knee. Only one case of BCG spondylitis was identified.

In response to the relatively high incidence of BCG osteomyelitis, the vaccination schedule for infants was officially changed in January 2016 from within 24 h after birth to between 5 and 8 months of age. This schedule change resulted in a significant decline in the incidence of BCG osteomyelitis from 41.4 to 7.9 events per million doses between the 2014–2015 and 2016–2018 periods (*p* = 0.0014, Table 2). However, administering the vaccine in older infants led to a substantial increase in incidences of abscesses (from 17.1 to 118.5 events per million doses, *p* < 0.0001) and lymphadenitis (from 34.1 to 61.2 events per million doses, *p* = 0.0724). Lymphadenitis tended to occur sooner post immunization in infants vaccinated at 5–8 months compared to those vaccinated within 24 h of birth (*p* < 0.0001, Table 2).

### 3.2. Influenza Vaccine

The most common ARs for influenza vaccines were neurological disorders (28.2%), allergies (28.2%), and rheumatological disorders (17.9%) (Table 3 and Figure 3). The time interval from vaccination to disease onset varied greatly among these ARs. Most allergic reactions occurred within one day of vaccination (median < 1 day, range < 1 day to 60 days, Table 3), whereas neurological disorders had a significantly longer onset interval, with a mean and median duration of 39.9 and 6.0 days, respectively (Table 3).

Guillain–Barré syndrome (GBS) was the most frequently observed neurological AR (6 events, 54.5%), with an incidence of 0.26 events per million influenza vaccine doses. GBS predominantly affected adult vaccinees over 40 years of age and typically presented within two weeks post vaccination. Idiopathic thrombocytopenic purpura (ITP) occurred in both pediatric and adult vaccinees. Urticaria generally developed within one day of vaccination. Notably, one pediatric vaccinee experienced anaphylactic shock on the day of vaccination. There was one case of mortality linked to GBS, adjudicated as “indeterminate”.

### 3.3. Pneumococcal Vaccine

Taiwan routinely administers two pneumococcal vaccines: the 13-valent pneumococcal conjugate vaccine (PCV13) for young children and the 23-valent pneumococcal polysaccharide vaccine (PPV23) for the elderly and individuals with underlying conditions [15]. The most common AR for both PCV13 (14 events, 60.9%) and PPV23 (6 events, 85.7%) was a severe local reaction, which typically occurred rapidly, with most reactions happening within one day post vaccination (69.6% for PCV13 and 85.7% for PPV23). Other less frequent ARs for PCV13 included urticaria (three events, 13.0%) and idiopathic thrombocytopenic purpura (ITP) (three events, 13.0%). Notably, severe local reactions presenting as extensive limb swelling were identified in two cases (28.6%) involving the PPV23 vaccine.

### 3.4. Japanese Encephalitis (JE) Vaccine

Since May 2017, Taiwan has replaced the mouse brain-derived JE vaccine with the live attenuated chimeric virus JE vaccine (Imojev^®^, Sanofi Pasteur, Lyon, France) and the inactivated Vero cell culture-derived JE vaccine (Jeval^®^, Adimmune, Taichung, Taiwan). These newer vaccines were associated with one adverse reaction case involving a skin rash after using the attenuated chimeric virus vaccine. In contrast, among the users of the older mouse brain-derived JE vaccine, four cases exhibited urticaria or skin rash, and there was one anaphylactic shock incident. A significant case involved a 5-year-old girl who experienced loss of consciousness within five minutes of receiving both the mouse brain-derived JE vaccine and the MMR vaccine. She was adjudicated as an “indeterminate” case, possibly due to anaphylactic shock or an immunization stress-related response.

### 3.5. Other EPI Vaccines

Table 4 outlines the incidences and characteristics of ARs related to other EPI vaccines. Notable ARs included the development of varicella, herpes zoster, and urticaria in a varicella vaccinee. For the DTaP-Hib-IPV or DTaP-IPV-Hib-HepB combination vaccines, the predominant ARs were severe local reactions (52.9%), cellulitis (17.6%), and ITP (17.6%). ITP was also the leading AR for the MMR vaccine, occurring in 9 out of 16 cases (56.3%), with a symptom onset interval ranging from 6 to 24 days post vaccination (mean of 14.2 days). One case involved optic neuritis in an HPV vaccinee, deemed “indeterminate” in relation to the HPV vaccine. A noteworthy incident involved a one-month-old boy who suffered from thrombocytopenia and accompanying intracranial hemorrhage nine days after HBV immunization, adjudicated as “indeterminate”. Another case of the simultaneous administration of the hepatitis A vaccine (HAV) and the influenza vaccine led to anaphylactic shock, adjudicated as “indeterminate”.

### 3.6. Claims of Anaphylactic Shock and Mortality

Anaphylactic shock was reported in three cases: one associated with the influenza vaccine, the other involving the simultaneous administration of the influenza and HAV vaccines, and a third in conjunction with the JE and MMR vaccines. In the latter two cases, immunization stress-related responses were also considered possible diagnoses. All these individuals survived, although data on their long-term follow-up were unavailable.

During the study period, there were 24 claims related to deaths (10 involving influenza, 3 rotavirus, 3 DTaP-Hib-IPV/DTaP-IPV-Hib-HepB, and 2 HepB) and 4 claims for life-threatening events (BCG and HepB, 2 each). Except for 1 fatality related to influenza vaccine-associated GBS, the remaining 23 death claims were adjudicated as “unrelated” to the vaccines. Notably, there were no claims involving congenital anomalies or birth defects.

## 4. Discussion

The BCG vaccine has been extensively used in regions where tuberculosis is prevalent and has shown optimal protection against severe manifestations such as meningitis and disseminated disease [16]. This study found that, among the EPI vaccines, BCG was associated with the highest incidence of ARs. In 2007, active surveillance by the Taiwan CDC led to a marked increase in the reported BCG-related AEFIs, particularly osteitis and osteomyelitis, with incidence rates rising approximately tenfold from 3.68 to 30.1 per million doses between the 2002–2006 and 2008–2012 periods [17,18]. Osteomyelitis symptoms typically emerged 6 to 12 months post vaccination, with diagnosis taking an average of 16.4 months [19]. Fortunately, BCG osteomyelitis generally has a good prognosis, with long-term sequelae being rare following adequate anti-tuberculosis therapy [19,20,21].

The relatively high incidence of BCG osteomyelitis in Taiwan prompted a change in the national immunization policy in 2016, delaying BCG administration from 24 h after birth to 5 to 8 months of age. Huang et al. observed a decrease in the incidence of BCG-related osteitis/osteomyelitis and an increasing rate of regional lymphadenitis and injection site reactions in the birth cohort from 2012 to 2017 following this policy change [22]. A more mature and robust immune response against BCG strains may account for this observation. Our study supports these findings, noting approximately a fivefold reduction in BCG-related osteitis and osteomyelitis while the timing after vaccination and the affected skeletal regions remained consistent. In Japan, the recommended age of BCG vaccination was less than 6 months after 2005 but subsequently changed to 5–8 months (latest amendment, <1 year) in 2013. National surveillance from 2013 to 2017 demonstrated that the incidence of developing BCG-associated osteitis/osteomyelitis and disseminated BCG was higher among those who received the BCG vaccine at <6 months of age than among those who were 6 months or older [23]. These data suggest that delaying BCG vaccination could reduce the incidence of these serious bone infections. However, potential disadvantages include the uncertainty of increased severe tuberculosis cases during the vaccination delay window, warranting further surveillance in infants under 5 months. Also, abscesses and lymphadenitis became more frequent with an older vaccination age, developing sooner post vaccination, causing prolonged and troublesome symptoms for caregivers. In response, the Taiwan CDC issued guidelines to better manage these ARs [24].

Along with the policy change in the vaccination age for BCG, the manufacturer of the BCG vaccines temporarily changed from the National Health Research Institutes (NHRI) in Taiwan to the Japan BCG Laboratory© between July 2016 and August 2020 due to the shortage in stockpile. However, both vaccines contained the same Tokyo-172 strains, fulfilling the WHO requirements. The intradermal route of administration did not change either. The manufacturer was changed back to the NHRI after August 2020. Continuing observation of the incidences of BCG abscess and osteomyelitis will be mandatory to clarify the role of different vaccine manufacturers in the occurrence of ARs of the BCG vaccine. Like the finding in this study, BCG was also the vaccine most frequently compensated for ARs in many other countries [25,26]. However, the estimated incidences of AEs following BCG vaccination varied greatly in different countries and ranged from 3.4 to 1540 per million vaccine doses [27,28,29]. Of note, an investigation into the AEs following BCG immunization from2013 to 2018 in Korea indicated that the incidence was as high as 260 per million vaccine doses [30]. Consistent with our findings, abscesses and lymphadenitis were the major disease entities caused by BCG in Australia, Korea, Poland, and Oman [26,28,30].

Although the influenza vaccine was the second most frequent in AR-related claims, its overall AR incidence was extremely low. Severe ARs like neurological disorders, anaphylactic shock, and ITP were observed at rates below one per million doses, underscoring the influenza vaccine’s excellent safety profile in post-marketing surveillance. Given its widespread use across global populations, injury claims remain common across regions and demographics [25,31], with a notable spike in 2010 due to extensive social media coverage during the 2009 H1N1 pandemic [7]. The low AR incidence data from this study are crucial for bolstering public trust and encouraging influenza vaccine uptake.

Guillain–Barré syndrome (GBS) is historically considered a significant AR of influenza vaccines, linked notably to the 1976 swine-origin influenza vaccine [32,33,34]. There remain public concerns regarding GBS and its association with influenza vaccination, notably during the 2009 H1N1 pandemic [35,36], although recent meta-analyses have not identified such associations [37]. Influenza vaccine-associated GBS is viewed as an immune-mediated condition, generally manifesting within six weeks post vaccination [38]. This study observed GBS occurrence within two weeks post immunization, reinforcing the notion that the benefits of influenza vaccination considerably outweigh potential AR risks.

Based on the findings of this study, it is recommended that Taiwan’s health authorities consider further refining the immunization schedule and public education regarding vaccine safety. Specifically, the change in the BCG vaccination age from immediately after birth to 5–8 months has resulted in a significant reduction in cases of osteomyelitis, suggesting that the postponement of the BCG vaccine may be beneficial in mitigating severe adverse reactions. However, this policy change has been associated with an increase in reported abscesses and lymphadenitis, indicating the need for healthcare professionals to be vigilant in monitoring and managing these conditions. It is also imperative to maintain and enhance educational campaigns that reiterate the overall safety and importance of vaccines, as the low incidence of serious adverse reactions further supports the continued use of vaccines such as the influenza vaccine, which demonstrated excellent safety despite its common use. These educational efforts could help alleviate vaccine hesitancy by ensuring that the public is well-informed about the benefits of vaccination versus the low risks of adverse effects. Additionally, continued enhancement of the VICP to ensure transparency and accessibility may further boost public confidence in the nation’s immunization activities.

## 5. Limitations

This study was based on a statistical analysis of the VICP database. There were some limitations regarding the system. First, identifying the specific vaccine responsible for systemic adverse reactions was challenging, particularly when multiple vaccines were administered simultaneously or in combination. Second, there was significant variability in the reported adverse event entities, which lacked a consistent form of reporting and a standardized terminology, potentially leading to misclassification. For example, severe local reactions with extensive limb swelling were often reported as cellulitis, which might not accurately reflect the nature of the reaction. Third, while adjudications were largely based on the available scientific evidence, there were no established quality criteria for the evidence used by the expert committee. In complex cases, this could result in subjective adjudications or reliance on evidence of a lower quality. Fourth, because the VICP operates as a passive reporting system, which is not mandatory, there was a potential for the underreporting of adverse reactions, particularly mild ones. However, the Taiwanese government encourages reporting through the VICP, supported by the possibility of financial compensation, so adverse reactions of significant severity and clinical importance were likely captured. Lastly, the time limitations for claim submission—either within two years of awareness of an adverse event or within five years of symptom onset—make it challenging to accurately estimate incidences of adverse events with long intervals post immunization, such as BCG osteomyelitis. Moreover, as VICP cases are closed once the final causality is determined and the financial compensation is made, information on long-term sequelae, in terms of both the physical and psychosocial aspects, is lacking.

## 6. Conclusions

Our study provides a comprehensive analysis of the adverse reactions associated with vaccines included in Taiwan’s EPI using data from the VICP from 2014 to 2019. The findings confirm the overall safety of these vaccines, with most adverse events occurring at low incidence rates. Notably, the BCG vaccine, despite its effectiveness in preventing severe forms of tuberculosis, showed the highest incidence of adverse reactions, predominantly due to subacute or chronic infections by BCG strains, such as osteomyelitis. This underscores the need for carefully evaluating the risk–benefit ratio in regions with low tuberculosis prevalence, where routine BCG vaccination may require re-assessment of its necessity versus potential adverse effects.

While the influenza vaccine was the second vaccine most frequently associated with adverse claims, it maintained an excellent safety profile with very low incidences of severe adverse reactions such as Guillain–Barré syndrome. This reinforces its continued use and importance in public health efforts to prevent influenza outbreaks.

Our analysis highlights the critical role of continuous post-marketing surveillance and compensation programs like the VICP in ensuring vaccine safety and maintaining public trust. These programs are invaluable for identifying rare adverse events and facilitating data-driven adjustments in immunization schedules, such as the adjusted timing of the BCG vaccine, which significantly reduced the incidence of severe reactions. Future research and policies should focus on optimizing vaccination strategies, potentially incorporating more personalized approaches to mitigate risks while maximizing the benefits of immunization across diverse populations.

## Figures and Tables

**Figure 1 vaccines-12-01133-f001:**
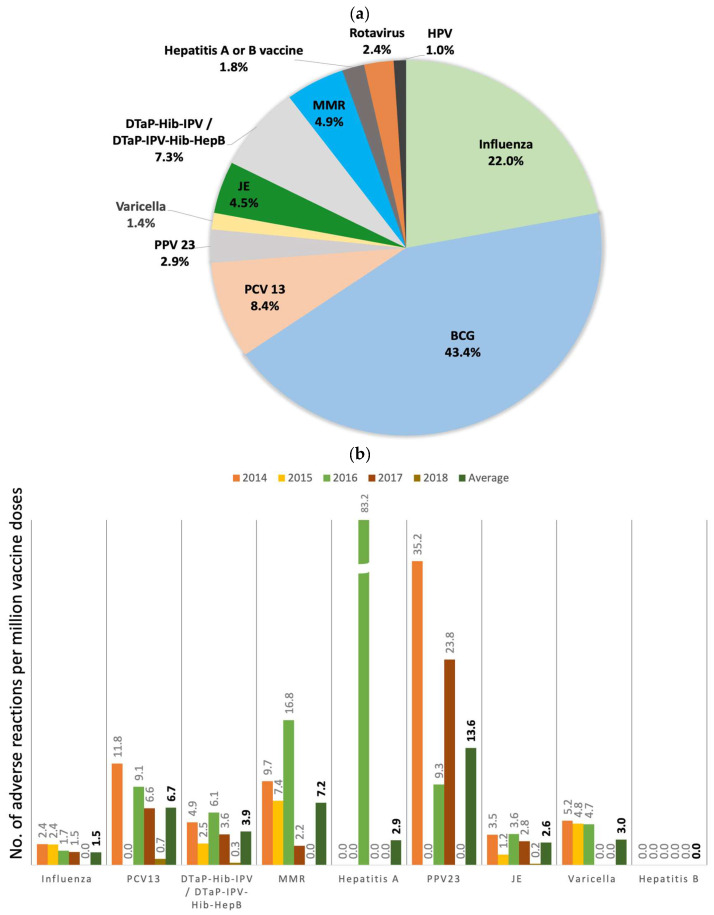
(**a**) Distribution of alleged vaccines in the claims received by the national vaccine injury compensation program in Taiwan during 2014 and 2019. Abbreviations: BCG, Bacillus Calmette–Guérin; PCV13, 13-valent pneumococcal conjugate vaccine; PPV23, 23-valent polysaccharide pneumococcal vaccine; JE, Japanese encephalitis vaccine; DTaP-Hib-IPV, diphtheria, tetanus, acellular pertussis, Hemophilus influenzae type b, and inactivated poliovirus vaccine; DTaP-IPV-Hib-HepB, diphtheria, tetanus, acellular pertussis, Hemophilus influenzae type b, inactivated poliovirus, and hepatitis B; MMR, measles–mumps–rubella vaccine; and HPV, human papillomavirus vaccine. (**b**,**c**) The incidence of the adverse reactions per million doses of vaccines. The average incidence is indicated on the right column with bold letters.

**Figure 2 vaccines-12-01133-f002:**
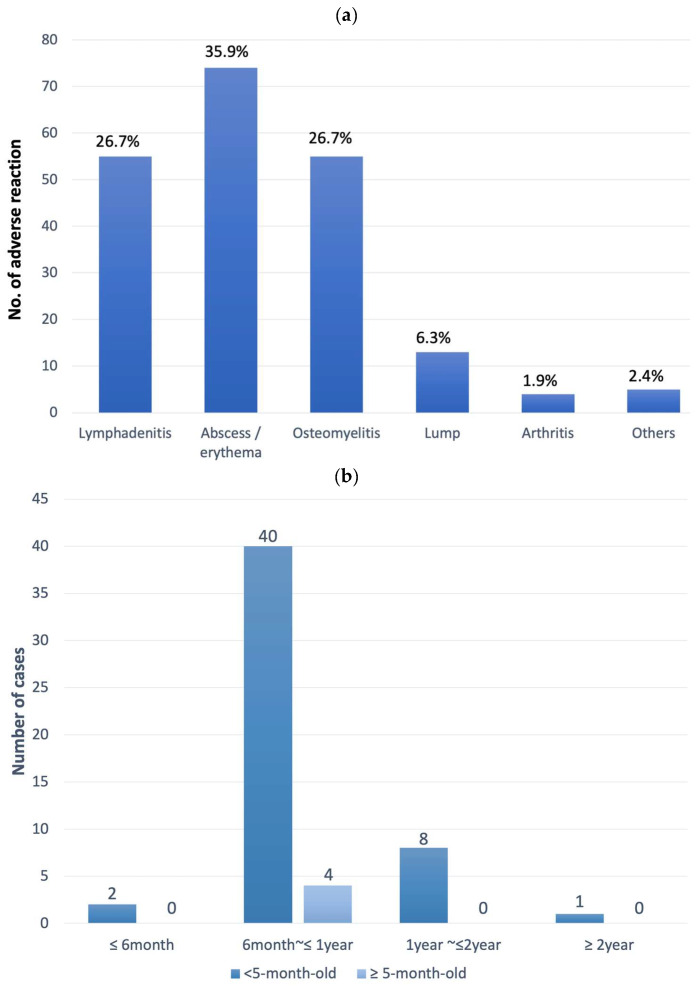
(**a**) The adverse reactions of the BCG vaccine. The percentages are the proportions of the indicated adverse reactions among all BCG-related adverse reactions. (**b**) The distributions of the intervals between the onset of the symptoms of osteomyelitis and BCG immunization stratified by the age of immunization.

**Figure 3 vaccines-12-01133-f003:**
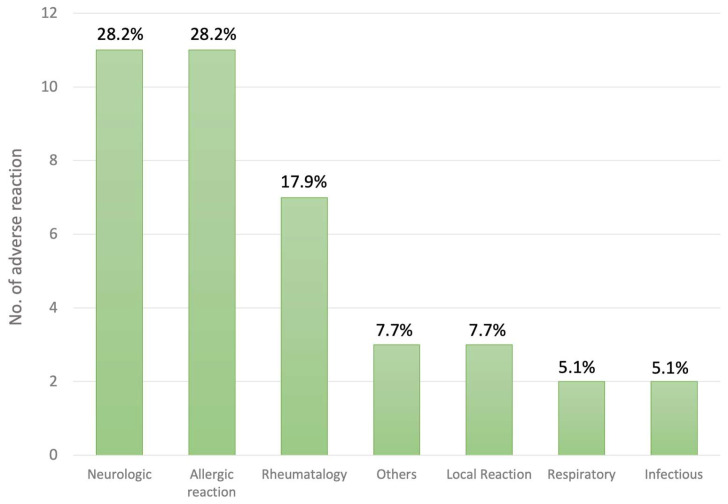
The adverse reactions of the influenza vaccine. The percentages indicate the proportion of adverse reactions.

**Table 1 vaccines-12-01133-t001:** Claims of vaccine injury and adjudication results in the Taiwan vaccine injury compensation program from 2014 to 2019.

Vaccine Type	No. of Events (%)			
	Adverse Reactions			Unrelated Events
	Related	Indeterminate	Total	
BCG vaccine (*n* = 213)	188 (88.3)	18 (8.5)	206 (96.7)	7 (3.3)
Influenza vaccine (*n* = 108)	4 (3.7)	35 (32.4)	39 (36.1)	69 (63.9)
PCV13 (*n* = 41)	14 (34.1)	9 (22.0)	23 (56.1)	18 (43.9)
DTaP-Hib-IPV/DTaP-IPV-Hib-HepB vaccine (*n* = 36)	12 (33.3)	5 (13.9)	17 (47.2)	19 (52.8)
MMR vaccine (*n* = 24)	1 (4.2)	15 (62.5)	16 (66.7)	8 (33.3)
JE vaccine (*n* = 22)	5 (22.7)	5 (22.7)	10 (45.5)	12 (54.5)
PPV23 (*n* = 14)	7 (50.0)	0 (0)	7 (50.0)	7 (50.0)
Rotavirus vaccine (*n* = 12)	0 (0)	3 ^a^ (25.0)	3 (25.0)	9 (75.0)
Hepatitis A or B vaccine (*n* = 9)	0 (0)	2 (22.2)	2 (22.2)	7 (77.8)
Varicella vaccine (*n* = 7)	0 (0)	3 (42.9)	3 (42.9)	4 (57.1)
HPV vaccine (*n* = 5)	0 (0)	1 (20.0)	1 (20.0)	4 (80.0)
Total claims (*n* = 491)	231 (47.0)	96 (19.6)	327 (66.6)	164 (33.4)

^a^ All three ARs involved intussusception. Abbreviations: BCG, Bacillus Calmette–Guérin; PCV13, 13-valent pneumococcal conjugate vaccine; PPV23, 23-valent polysaccharide pneumococcal vaccine; JE, Japanese encephalitis vaccine; DTaP-Hib-IPV, diphtheria, tetanus, acellular pertussis, *Hemophilus influenzae* type b, and inactivated poliovirus vaccine; DTaP-IPV-Hib-HepB, diphtheria, tetanus, acellular pertussis, *Hemophilus influenzae* type b, inactivated poliovirus, and hepatitis B; MMR, measles–mumps–rubella; and HPV, human papillomavirus.

**Table 2 vaccines-12-01133-t002:** Post-immunization incidence and duration of various adverse reactions (ARs) of the BCG vaccine at different immunization ages, in 2014–2018.

Adverse Reactions/Anatomic Sites	No. of AR (Incidence per Million Doses)				Months between Immunization and AR Onset, Median (Range)			
	Overall *n =* 917,030 Vaccine Doses	Age of Immunization			Overall *n* = 917,030 Vaccine Doses	Age of Immunization		
		<5 Months *n =* 410,621	≥5 Months *n* = 506,409	*p*		<5 Months *n* = 410,621	≥5 Months *n* = 506,409	*p*
Abscess	67 (73.1)	7 (17.1)	60 (118.5)	<0.0001	2.0 (0.6–14.9)	4.9 (0.6–10.2)	2.0 (0.6–14.9)	0.0195
Left arm	64	6	58		2.1 (0.6–14.9)			
Chest wall	1	1	0		7.2			
Left axillary	2	0	2		1.3 (1.1–1.5)			
Lymphadenitis	45 (49.1)	14 (34.1)	31 (61.2)	0.0724	1.8 (0.9–14.6)	4.7 (1.4–14.6)	1.6 (0.9–4.8)	<0.0001
Left axillary	34	11	23		2.0 (0.9–14.6)			
Not specified	8	2	6		1.5 (1.0–3.3)			
Clavicles	3	1	2		1.5 (1.4–1.7)			
Osteomyelitis	21 (22.9)	17 (41.4)	4 (7.9)	0.0014	14.5 (9.0–33.8)	14.5 (9.0–33.8)	14.3 (9.6–15.4)	0.4705
Lower limbs Long bone	10	6	4		15.4 (9.6–22.1)			
Sternum/rib	4	4	0		11.5 (9.6–13.7)			
Lower limb Foot, ankle, knee	3	3	0		12.1 (9.0–33.8)			
Unknown	2	2	0		18.4 (14.5–22.2)			
Upper limb Long bone	2	2	0		22.4 (22.0–22.9)			
Lumps	8 (8.7)	7 (17.0)	1 (2.0)	0.0262	10.4 (1.2–19.2)	14.1 (1.2–19.2)	2.5 (2.0–3.1)	…
Chest wall	7	6	1		14.1 (3.1–19.2)			
Four limbs	1	1	0		1.2			
Arthritis	3 (3.3)	1 (2.4)	2 (3.9)	>0.9999	14.0 (5.6–25.0)	14.0	15.3 (5.6–25.0)	…
Others ^a^	5 (5.5)	1 (2.4)	4 (7.9)	0.3886	0.9 (<1d–14.9)	0.46	0.7 (<1d–1.5)	…

^a^ Others: dermatitis (1), granuloma (1), urticaria (1), lupus vulgaris (1), and immunization stress-related response (1).

**Table 3 vaccines-12-01133-t003:** Incidences and demographics of various adverse reactions (ARs) of the influenza vaccine and intervals between immunization and the onset of the AR.

Categories of Adverse Reactions/Adverse Reactions	No. of ARs (Incidence per Million Doses)	Male (%)	Ages of Immunization in Years, Mean ± Standard Deviation	Days between Immunization and AR Onset, Mean ± Standard Deviation (Median, Range)
Neurology	11 (0.47)	6 (54.5)	47.9 ± 25.6	39.9 ± 108.0 (6.0, 1–365)
Guillain–Barré syndrome	6 (0.26)	4 (66.7)	52.2 ± 24.9	6.8 ± 5.6 (6.5, 1–14)
Transverse myelitis	2 (0.09)	1 (50.0)	41.5 ± 19.1	12.0 ± 14.1 (12, 2–22)
Acute myelitis	1 (0.04)	0 (0.0)	72.0	365.0
Acute disseminated encephalomyelitis	1 (0.04)	0 (0.0)	1.0	1.0
Tolosa–Hunt syndrome	1 (0.04)	1 (100.0)	58.0	6.0
Allergic reaction	11 (0.47)	4 (36.4)	26.2 ± 22.6	6.2 ± 17.9 (<1, <1–60)
Urticaria	10 (0.43)	4 (40.0)	28.5 ± 22.4	6.8 ± 18.7 (<1, <1–60)
Anaphylactic shock	1 (0.04)	0 (0.0)	3.0	<1.0
Rheumatology	7 (0.30)	3 (42.9)	26.7 ± 29.2	5.2 ± 5.4 (4.0, 0.5–17)
Idiopathic thrombocytopenic purpura	6 (0.26)	3 (50.0)	23.8 ± 28.6	6.0 ± 5.4 (4, 3–17)
Vasculitis	1 (0.04)	0 (0.0)	52.0	<1.0
Local reaction	3 (0.13)	2 (66.7)	4.3 ± 0.6	<1.0 (0.5, 0.5)
Respiratory	2 (0.09)	1 (50.0)	39.0 ± 11.3	<1.0 (0.5, 0.5)
Infectious	2 (0.09)	1 (50.0)	61.5 ± 10.6	1.5 ± 0.7 (1.5, 1–2)
Others ^a^	3 (0.13)	2 (66.7)	14.3 ± 5.5	1.7 ± 2.0 (0.5, 0.5–4)
Total	39 (1.68)	19 (48.7)	32.3 ± 25.9	32.9 ± 97.7 (2, 0.5–365)

^a^ Others: lipoatrophy (1), oral ulcer (1), and limb and chest pain (1).

**Table 4 vaccines-12-01133-t004:** Incidence and demographics of various adverse reactions and intervals between immunization and onset of ARs of selected EPI vaccines.

Vaccine Types	Adverse Reactions	No. of ARs (Incidence per Million Doses)	Gender Male (%)	Ages of Immunization in Years, Mean (Range)	Days between Immunization and AR Onset, Mean ± Standard Deviation (Median, Range)
PCV13 (*n* = 23)	Severe local reaction	14 (4.3)	7 (50.0)	15.5 (0.3–70.0)	1.4 ± 1.6 (1, 0.5–5)
	Urticaria	3 (0.9)	3 (100.0)	0.8 (0.3–1.0)	1.3 ± 1.4 (0.5, 0.5–3)
	ITP	3 (0.9)	1 (33.3)	21.8 (0.2–65.0)	17.7 ± 11.9 (23, 4–26)
	Cellulitis	2 (0.6)	0 (0.0)	31.0 (1.0–61)	<1.0 (0.5, 0.5–0.5)
	Intussusception	1 (0.3)	1 (100.0)	0.3	3.0
DtaP-Hib-IPV/DtaP-IPV-Hib-HepB (*n* = 17)	Severe local reaction	9 (2.3)	7 (77.8)	1.4 (0.3–2.0)	0.9 ± 0.5 (1.0, 0.5–2.0)
	ITP	3 (0.8)	1 (33.0)	1.5 (0.2–1.0)	19.0 ± 15.7 (26, 1–30)
	Cellulitis	3 (0.8)	2 (66.7)	2.0	1.2 ± 0.8 (1.0, 0.5–2.0)
	Seizure	1 (0.2)	0 (0.0)	0.2	<1.0
	Urticaria	1 (0.2)	1 (100.0)	0.3	<1.0
MMR (*n* = 16)	ITP	9 (4.3)	6 (66.7)	1.6 (1.0–6.0)	14.2 ± 6.9 (14.0, 6.0–24.0)
	Urticaria	4 (1.9)	2 (50.0)	3.3 (1.0–5.0)	1.1 ± 1.3 (0.5, 0.5–3.0)
	Cellulitis	1 (0.5)	0 (0.0)	1.0	3.0
	Anaphylactic shock	1 (0.5)	0 (0.0)	5.0	<1.0
	Erythema multiforme	1 (0.5)	1 (100.0)	1.0	4.0
JE (*n* = 10)	Urticaria	5 (1.3)	4 (80.0)	1.2 (1.0–2.0)	1.7 ± 1.9 (1.0, 0.5–5.0)
	Severe local reaction	3 (0.5)	2 (66.7)	2.0 (2.0–2.0)	0.8 ± 0.3 (1.0, 0.5–1.0)
	Anaphylactic shock	1 (0.3)	0 (0.0)	5.0	<1.0
	Cellulitis	1 (0.3)	1 (100.0)	2.0	2.0
PPV23 (*n* = 7)	Severe local reaction	6 (11.6)	1 (14.2)	68.5 (62.0–75.0)	5.7 ± 11.9 (1, 0.5–30)
	Abscess	1 (1.9)	1 (100.0)	76.0	<1.0
VZV (*n* = 3)	Varicella and herpes zoster	1 (1.0)	1 (100.0)	1.0	60.0
	Varicella	1 (1.0)	1 (100.0)	1.0	6.0
	Urticaria	1 (1.0)	1 (100.0)	1.0	3.0
Hepatitis A or B (*n* = 2)	Anaphylactic shock	1 (2.9)	1 (100.0)	0.1	9.0
	ITP	1a	0 (0.0)	3.0	<1.0
HPV vaccine (*n* = 1)	Optic neuritis	1b	0 (0.0)	39.0	30.0

## Data Availability

The raw data of this study can be obtained by contacting the corresponding author.

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
