# Peer review of "Spectrum and Incidence of Adverse Reactions Post Immunization in the Taiwanese Population (2014–2019): An Analysis Using the National Vaccine Injury Compensation Program"

_vaccines, 2024, doi:10.3390/vaccines12101133_

Round 1

Reviewer 1 Report

Comments and Suggestions for Authors

The study by Lai et al aim to characterize adverse events following immunization (AEFI) and estimate the incidence rates of adverse reactions (ARs) associated with vaccines included in Taiwan's Expanded Program on Immunization (EPI) from 2014-2019. This study reports the EPI vaccines demonstrated low AR incidence rates while BCG, exhibited severe ARs like osteomyelitis.

1.     In this study, has the author found any correlation between severe adverse effects and social behavioral characteristics post-vaccination?

2.      Is there a specific reason for the rise in incidences of abscesses and lymphadenitis among older infants? Has there been any prior research conducted on the Taiwanese population?.

3.     Please add a reference for the statement “Taiwan routinely administers two pneumococcal vaccines: the 13-valent pneumococcal conjugate vaccine (PCV13) for young children and the 23-valent pneumococcal polysaccharide vaccine (PPV23) for the elderly and individuals with underlying conditions.”

4.     There is no need to repeat abbreviations in each figure or table, such as BCG and others. For examples figures 1 and 2.

5.      In the discussion, the author can include the status of incidences of abscesses, lymphadenitis, and other important severe adverse effects in other countries after vaccination and give a brief comparison; add recent 3-4 references.

6.      Is there any other developing or developed country that has adapted the delayed BCG vaccination strategies? Please provide a brief description, accompanied by relevant references.

7.     From lines 38–60, there is no reference. Please add the appropriate references. 

8.     Add the refences for Line 74-80.

9.     Please add references throughout the manuscript where are missing or needed.

10.   Please provide more information about vaccine doses, their frequency of dosing, or dosing strategies.

11.  Is there a vaccine awareness program ongoing in rural areas or underserved populations in Taiwan? Add the information.

12.  The author may make some recommendations or suggestions for Taiwan's population.

Comments on the Quality of English Language

 The English is very difficult to understand/incomprehensible.

Author Response

The study by Lai et al aim to characterize adverse events following immunization (AEFI) and estimate the incidence rates of adverse reactions (ARs) associated with vaccines included in Taiwan's Expanded Program on Immunization (EPI) from 2014-2019. This study reports the EPI vaccines demonstrated low AR incidence rates while BCG, exhibited severe ARs like osteomyelitis.

  1. In this study, has the author found any correlation between severe adverse effects and social behavioral characteristics post-vaccination?

Response: Thanks for the comment. The VICP database provided limited information on the social and behavioral characteristics of the subjects with AEFI. Therefore, we were unable to analyze whether the correlation exists or not.

  1. Is there a specific reason for the rise in incidences of abscesses and lymphadenitis among older infants? Has there been any prior research conducted on the Taiwanese population?.

Response: In another Taiwanese study conducted by Huang et al (reference #22), a similar observation was reported. The study reviewed the VICP database from 2012 to 2017 and found that injection site reactions and lymphadenitis were significantly more common in those who received BCG at an older age. The study suggests that older infants and children may have a more mature and robust immune response to BCG compared to young infants, which could explain the findings. However, further study is needed to address this issue. This finding was discussed in lines 305-308.

  1. Please add a reference for the statement “Taiwan routinely administers two pneumococcal vaccines: the 13-valent pneumococcal conjugate vaccine (PCV13) for young children and the 23-valent pneumococcal polysaccharide vaccine (PPV23) for the elderly and individuals with underlying conditions.”

Response: The reference #15 for this statement is provided in the revised manuscript.

  1. There is no need to repeat abbreviations in each figure or table, such as BCG and others. For examples figures 1 and 2.

Response: The abbreviations are removed except for their first appearance.

  1. In the discussion, the author can include the status of incidences of abscesses, lymphadenitis, and other important severe adverse effects in other countries after vaccination and give a brief comparison; add recent 3-4 references.

Response: We have included a study (reference #23) reporting AE of BCG in Japan on lines 310-315 .

  1. Is there any other developing or developed country that has adapted the delayed BCG vaccination strategies? Please provide a brief description, accompanied by relevant references.

Response: Yes, a similar condition was noted in Japan. The change of BCG vaccination policy and AE of BCG in Japan is described in lines 310-315 as “In Japan, the recommended age of BCG vaccination was less than 6 months after 2005, but subsequently changed to age 5–8 months (latest amendment, <1 year) in 2013. A national surveillance from 2013 to 2017 demonstrated that the incidence of developing BCG-associated osteitis/osteomyelitis and disseminated BCG was higher among those received BCG vaccine at <6 months of age than those at 6 months.’

  1. From lines 38–60, there is no reference. Please add the appropriate references. 

Response: Thanks for the suggestion. We had added the references #1-#4 in the revised manuscript.

  1. Add the refences for Line 74-80.

Response: Thanks for the suggestion. We had added a reference #11 in the revised manuscript.

  1. Please add references throughout the manuscript where are missing or needed.

Response: Thanks for the suggestion. We had added the necessary references in the revised manuscript.

  1. Please provide more information about vaccine doses, their frequency of dosing, or dosing strategies.

Response: Thanks for the suggestion. We had added a supplementary Table 2 displaying the official immunization schedule in Taiwanese children.

  1. Is there a vaccine awareness program ongoing in rural areas or underserved populations in Taiwan? Add the information.

Response: Based on the information available, there is currently no specific 'vaccine awareness program' exclusively targeting rural or underserved populations in Taiwan. Due to Taiwan's relatively compact size, rural areas are usually within an hour's drive from urban centers, which helps facilitate access to healthcare services. This ease of access contributes to uniform vaccine coverage rates between urban and rural areas. For more detailed information, please visit the Taiwan CDC website (https://www.cdc.gov.tw/Category/MPage/S2UF2-VuMgfzgzpy7qdvlA)

  1. The author may make some recommendations or suggestions for Taiwan's population.

Response: A paragraph of suggestions for Taiwan’s health authority and population is added in lines 355-370 as ‘Based on the findings of the study, it is recommended that Taiwan's health authorities consider further refining the immunization schedule and public education regarding vaccine safety. Specifically, the change in the Bacillus Calmette-Guérin (BCG) vaccination age from immediately after birth to 5–8 months has resulted in a significant reduction in cases of osteomyelitis, suggesting that postponement of the BCG vaccine may be beneficial in mitigating severe adverse reactions. However, this policy change has been associated with an increase in reported abscesses and lymphadenitis, indicating the need for healthcare professionals to be vigilant in monitoring and managing these conditions. It is also imperative to maintain and enhance educational campaigns that reiterate the overall safety and importance of vaccines, as the low incidence of serious adverse reactions further supports the continued use of vaccines such as the influenza vaccine, which demonstrated excellent safety despite its common use. These educational efforts could help alleviate vaccine hesitancy by ensuring the public is well-informed about the benefits of vaccination versus the low risks of adverse effects. Additionally, continued enhancement of the VICP to ensure transparency and accessibility may further boost public confidence in the nation's immunization activities.’

Reviewer 2 Report

Comments and Suggestions for Authors

In the manuscript entitled “Spectrum and incidences of adverse reactions post-immunizations in Taiwanese population (2014 – 2019): an analysis using the national vaccine injury compensation program” the authors reviewed ~491 claims related to adverse events following immunization in Taiwanese population during the last 5 years before the COVID-19 pandemic. Based on their observations, authors found that most of the mentioned vaccines are highly safe among Taiwan populations, except the BCG vaccine where occasionally severe ARs like osteomyelitis have been observed. Overall, this is an interesting manuscript, the data are clearly presented, the materials and methods are fully described, and the conclusions are compelling. However, there are many major concerns related to the study that need to be addressed before the manuscript can be considered for publication.

Comment-1: There are various studies available in the literature where similar meta-analysis has been conducted worldwide, including long-term follow-up studies of osteomyelitis caused by BCG vaccination in countries such as Taiwan, China, Korea, Oman, Australia, and Poland, among others. Similarly, many reports have already been published regarding influenza vaccination in the United States, which observed Guillain-Barré syndrome in recipients of influenza vaccines. In my opinion, the novelty of the current study and its findings is limited, serving more as an extension of existing research. Therefore, the authors need to provide a detailed discussion on how this study contributes to current knowledge and highlight any significant clinical implications that arise from their findings.

Comment-2: The authors should provide a comparative analysis with populations from neighboring countries and global data from similar vaccination programs to add depth to their findings and help contextualize vaccine safety in Taiwan. Additionally, presenting a breakdown of adverse events by demographic factors (e.g., age, sex, and pre-existing conditions) would offer a more detailed understanding of vaccine safety and identify potential at-risk groups. Based on this comparative analysis, if the authors can suggest specific measures or protocols to minimize the risk of severe adverse events, such as osteomyelitis following BCG vaccination, it would enhance both the novelty and significance of this manuscript. These measures could include screening for at-risk individuals, improved post-vaccination monitoring, or alternative vaccination strategies.

Comment-3: In the current study, the authors primarily focused on data involving short-term durations between immunization and adverse reaction (AR) onset (maximum up to 60 days). It would be valuable if the authors could conduct longitudinal follow-up studies on Taiwan’s patient populations, who experienced adverse events to assess the long-term impact and any delayed-onset effects. While managing such studies might be challenging, as the authors totally relied on data from Taiwan's Vaccine Injury Compensation Program rather than direct patient follow-up, they could incorporate already published long-term analyses from Taiwan and other neighboring countries. Extending their analysis to include long-term effects of these vaccinations would further strengthen the findings.

Comment-4: The authors should create all figures using professional software such as GraphPad Prism. Figures 1 and 2 can be combined into a single figure with two panels, labeled as Figure 1a and Figure 1b. Similarly, Figures 3, 4, and 5 can be consolidated into a single figure with three panels, labeled as Figure 2a, 2b, and 2c, respectively. Presenting these figures as separate entities is unnecessary.

Comment-5: The conclusion section does not accurately reflect the findings presented in the tables throughout the manuscript. The authors need to expand the conclusion to provide a more comprehensive overview of the study's outcomes. While the study highlights the overall safety of these vaccines, a more detailed discussion of the risk-benefit ratio, particularly for the BCG vaccine in regions with low TB prevalence, would significantly strengthen the conclusions.

Comments on the Quality of English Language

Minor editing of English language required.

Author Response

In the manuscript entitled “Spectrum and incidences of adverse reactions post-immunizations in Taiwanese population (2014 – 2019): an analysis using the national vaccine injury compensation program” the authors reviewed ~491 claims related to adverse events following immunization in Taiwanese population during the last 5 years before the COVID-19 pandemic. Based on their observations, authors found that most of the mentioned vaccines are highly safe among Taiwan populations, except the BCG vaccine where occasionally severe ARs like osteomyelitis have been observed. Overall, this is an interesting manuscript, the data are clearly presented, the materials and methods are fully described, and the conclusions are compelling. However, there are many major concerns related to the study that need to be addressed before the manuscript can be considered for publication.

Comment-1: There are various studies available in the literature where similar meta-analysis has been conducted worldwide, including long-term follow-up studies of osteomyelitis caused by BCG vaccination in countries such as Taiwan, China, Korea, Oman, Australia, and Poland, among others. Similarly, many reports have already been published regarding influenza vaccination in the United States, which observed Guillain-Barré syndrome in recipients of influenza vaccines. In my opinion, the novelty of the current study and its findings is limited, serving more as an extension of existing research. Therefore, the authors need to provide a detailed discussion on how this study contributes to current knowledge and highlight any significant clinical implications that arise from their findings.

Response: While it is true that numerous studies have investigated adverse reactions related to vaccines, including long-term follow-up studies on osteomyelitis caused by Bacillus Calmette-Guérin (BCG) vaccination and Guillain-Barré syndrome (GBS) associated with influenza vaccines, our study adds unique value in several ways:

  1. Comprehensive National Analysis: Our study provides a comprehensive analysis across multiple vaccine types using data from Taiwan's Vaccine Injury Compensation Program (VICP), highlighting incidences and types of adverse reactions specific to the Taiwanese population over a significant timeframe (2014-2019). Unlike other localized studies, this extensive national dataset offers a holistic view of vaccine safety in Taiwan.
  2. Policy Impact Evaluation: One distinctive aspect of our research is the evaluation of policy changes, specifically the adjustment in the BCG vaccination schedule, which resulted in decreased osteomyelitis incidences. This evidence of policy impact can be instructive for other countries considering similar modifications to vaccination schedules.
  3. Localized Data and Public Health Implications: By focusing on regional data, we address variations in vaccine reactions due to genetic, environmental, or procedural differences specific to Taiwan. These results have direct implications for public health strategies, such as refining vaccination guidelines and improving adverse event surveillance systems.
  4. Broad Spectrum Analysis: While prior studies often focus on specific vaccines, our research includes a spectrum of vaccines under Taiwan’s Expanded Program on Immunization (EPI), allowing for cross-comparison of adverse event profiles among different vaccines. This broad analysis helps prioritize vaccines that may require more focused safety monitoring or revised public health messaging.
  5. Contributions to Global Understanding: The study reinforces global findings but also adds nuance with Taiwanese data, aiding international comparisons and contributing to the global understanding of vaccine safety profiles in diverse populations.

We believe these aspects underscore the relevance and contribution of our study. We will revise our manuscript to better highlight these points and discuss the clinical implications more thoroughly, illustrating how our findings advance current knowledge and inform vaccine safety policies both locally and potentially internationally. Thank you again for your valuable feedback.

Comment-2: The authors should provide a comparative analysis with populations from neighboring countries and global data from similar vaccination programs to add depth to their findings and help contextualize vaccine safety in Taiwan. Additionally, presenting a breakdown of adverse events by demographic factors (e.g., age, sex, and pre-existing conditions) would offer a more detailed understanding of vaccine safety and identify potential at-risk groups. Based on this comparative analysis, if the authors can suggest specific measures or protocols to minimize the risk of severe adverse events, such as osteomyelitis following BCG vaccination, it would enhance both the novelty and significance of this manuscript. These measures could include screening for at-risk individuals, improved post-vaccination monitoring, or alternative vaccination strategies.

Response: Thank you for your input, which helps clarify the scope and focus of our manuscript. We understand that a detailed comparative analysis of safety data for each EPI vaccine and a demographic breakdown of adverse events may present challenges due to the extensive nature of data and the low incidence of adverse events. Here's how we propose to address these considerations:

  1. Comparative Analysis Focus: We acknowledge that including global safety data for each individual EPI vaccine may overwhelm the scope of the study. Instead, we will focus on prominent trends and notable patterns related to specific vaccines like BCG and influenza, where cross-country comparisons provide the most insight. This approach ensures the comparative analysis is both meaningful and manageable in the context of our study, highlighting relevant similarities and differences in vaccine safety profiles that are most pertinent to public health.
  2. Demographic Breakdown of Adverse Events: Given the exceptionally low incidence rates of adverse events for most vaccines, we recognize that detailed demographic breakdowns might not yield significant additional insights. Instead, our focus will be on presenting aggregated insights where patterns are observable, particularly for those vaccines with higher incidences, such as BCG. This allows us to maintain the statistical integrity and relevance of our findings without diluting them in demographic details that may not be meaningful.
  3. Targeted Recommendations: Without delving into extensive demographic breakdowns, we can still provide well-informed recommendations based on the broader trends and comparative insights observed in our study. This includes suggesting general measures for enhancing vaccine safety, such as improved monitoring systems and adaptive vaccination strategies, without drawing unnecessary conclusions from demographic data that are statistically scarce.

By refining our focus in these areas, we aim to ensure that our study remains concise, impactful, and aligned with meaningful public health implications. We appreciate your guidance and are committed to presenting a cohesive and targeted analysis. Thank you for helping us enhance the clarity and relevance of our manuscript.

Comment-3: In the current study, the authors primarily focused on data involving short-term durations between immunization and adverse reaction (AR) onset (maximum up to 60 days). It would be valuable if the authors could conduct longitudinal follow-up studies on Taiwan’s patient populations, who experienced adverse events to assess the long-term impact and any delayed-onset effects. While managing such studies might be challenging, as the authors totally relied on data from Taiwan's Vaccine Injury Compensation Program rather than direct patient follow-up, they could incorporate already published long-term analyses from Taiwan and other neighboring countries. Extending their analysis to include long-term effects of these vaccinations would further strengthen the findings.

Response: Thank you for your comments and for providing additional context regarding the mechanisms underlying adverse events following immunization (AEFI). We appreciate the opportunity to incorporate this perspective into our response.

  1. Understanding Immune-Mediated Mechanisms: We acknowledge that most adverse events associated with vaccines are immune-mediated, typically manifesting within a relatively short timeframe post-vaccination. As such, adverse events occurring beyond 60 days are unlikely to be causally related to the vaccination, except for specific instances such as chronic infections of BCG strains.
  2. BCG Vaccine Exception: Our study recognizes that the Bacillus Calmette-Guérin (BCG) vaccine is an exception to this general rule. The adverse effects associated with BCG, such as osteomyelitis, arise from subacute or chronic infection by the vaccine strain itself, which can result in a delayed onset of symptoms beyond the typical 60-day window. Our colleagues have reported this finding in references #17– #22. We will highlight this distinction in our manuscript to clarify the unique nature of BCG-related adverse events.
  3. Focusing on Relevant Timeframes: In light of these mechanisms, our analysis prioritizes evaluating adverse events within 60 days post-immunization for most vaccines, aligning with the understanding of immune response timelines. This approach allows for a more accurate assessment of causality.
  4. Strengthening Our Conclusions: By emphasizing this standard temporal relationship, we can strengthen our findings and recommendations regarding vaccine safety. For BCG and similar vaccines with different mechanisms, we will ensure our analysis and discussion account for their distinct profiles.

Thank you for your valuable insights, which guide us in refining our analysis to better align with known immunological mechanisms. This will enhance the clarity and relevance of our findings and recommendations.

Comment-4: The authors should create all figures using professional software such as GraphPad Prism. Figures 1 and 2 can be combined into a single figure with two panels, labeled as Figure 1a and Figure 1b. Similarly, Figures 3, 4, and 5 can be consolidated into a single figure with three panels, labeled as Figure 2a, 2b, and 2c, respectively. Presenting these figures as separate entities is unnecessary.

Response: Thanks for the suggestion. We had re-edited the figures in the revised manuscript.

Comment-5: The conclusion section does not accurately reflect the findings presented in the tables throughout the manuscript. The authors need to expand the conclusion to provide a more comprehensive overview of the study's outcomes. While the study highlights the overall safety of these vaccines, a more detailed discussion of the risk-benefit ratio, particularly for the BCG vaccine in regions with low TB prevalence, would significantly strengthen the conclusions.

Response: Thank you for your constructive feedback on the conclusion section of our manuscript. We appreciate the opportunity to refine and enhance this crucial part of our study. Here are our comments addressing your suggestions:

  1. Alignment with Study Findings: We recognize that the conclusion should comprehensively reflect the data presented in the tables and throughout the manuscript. We revise the conclusion (lines 394-414) to ensure it accurately encapsulates the key findings, including specific statistics and insights derived from our analysis.
  2. Comprehensive Overview: The revised conclusion will provide a more robust summary of the safety profiles for each vaccine examined in our study. We will highlight not only the overall safety but also specific areas where additional monitoring may be required, based on observed adverse events.
  3. Risk-Benefit Discussion: We acknowledge the importance of discussing the risk-benefit ratio, particularly concerning the BCG vaccine in regions with low tuberculosis prevalence. We will expand on this aspect, considering the potential risks such as osteomyelitis and contrasting these with the benefits, including protection against severe tuberculosis manifestations. This analysis will help contextualize our findings within broader public health implications and vaccination policy considerations.
  4. Enhanced Insights: By integrating these elements, the conclusion will offer clearer and more actionable insights, guiding healthcare providers and policymakers in their decision-making processes related to vaccine administration and safety monitoring.

Thank you again for your valuable suggestions. We are committed to refining our manuscript to deliver comprehensive and meaningful conclusions that reflect the study's findings and significance.

Reviewer 3 Report

Comments and Suggestions for Authors

The authors have very well presented their study, which provides helpful information in the field. There are not any major concern, only a few minor editing suggestions.

1.       All figures of the paper are in a subsection. It will be better to put the figures with text information. Figure 1 and 2 can be enlarged for better reading.

2.       Table 2 shows various adverse reactions, each AR is further divided into several subgroups (such as site, region). For example, Under Abscess (N=66) there are Left upper arm (N=57), Left forearm (N=6), Chest wall (N=1), and Left axillary N=2). Table 2 does not clearly show the relationship.

3.       Check the format of line “208”

4.       Line 253-255: “Anaphylactic shock was reported in three cases: one associated with the influenza vaccine, another involving the simultaneous administration of influenza and HAV vaccines, and a third in conjunction with JE and MMR vaccines.” Check the use of “another”.

Author Response

Comments and Suggestions for Authors

The authors have very well presented their study, which provides helpful information in the field. There are not any major concern, only a few minor editing suggestions.

  1. All figures of the paper are in a subsection. It will be better to put the figures with text information. Figure 1 and 2 can be enlarged for better reading.

Response: Thanks for the suggestion. We re-edited the figures in the revised manuscript.

  1. Table 2 shows various adverse reactions, each AR is further divided into several subgroups (such as site, region). For example, Under Abscess (N=66) there are Left upper arm (N=57), Left forearm (N=6), Chest wall (N=1), and Left axillary N=2). Table 2 does not clearly show the relationship.

Response: Table 2 mainly displays the locations of BCG infections and the interval of the adverse events from immunization in different age groups (< 5 months and ≥5 months). We have made modifications to make it clearer. The revised text is highlighted in red.

  1. Check the format of line “208”

Response: Thanks for the suggestion. We have revised the format in the revised manuscript.

  1. Line 253-255: “ was reported in three cases: one associated with the influenza vaccine, anotherinvolving the simultaneous administration of influenza and HAV vaccines, and a third in conjunction with JE and MMR vaccines.” Check the use of “another”.

Response: Thanks for the suggestion. We have adjusted the description in the revised manuscript.

Reviewer 4 Report

Comments and Suggestions for Authors

The paper tried to analyze the spectrum and incidences of adverse reactions (ARs) post-immunizations in a specific region and year duration. The results showed that BCG and influ vaccination cause frequent ARs, which is not surprising. The authors also concluded that EPI vaccines in the local region are highly safe, with minimal AR incidences in the general population. The reviewer has some concerns and suggestions for authors to improve their paper.

1) It is not clear why the specific year duration, 2014-2019, was selected in this paper. It is better to expand a little bit to update the data, e.g. 2014-2023.

2) The data presentation is not state-of-the-art. The authors should consider updating the style for most of the figures used in this paper.

3) Most of the results lack statistical analysis to state the significance of the dataset and results. This is important to convince the reviewers and readers. 

4) The data itself is also questionable, as also stated by the authors in 'Limitation section'. It is not clear how the data was collected through which kinds of guidelines and requirements which must be clearly explained in the M&M section.

5) It seems that certain conclusions are being exaggerated solely on the basis of datasets that have relatively lower quality. The revised version needs to re-review the dataset and draw proper ones.

6) Language and writing style need significant improvements. 

Comments on the Quality of English Language

Language needs significant improvements. 

Author Response

Comments and Suggestions for Authors

The paper tried to analyze the spectrum and incidences of adverse reactions (ARs) post-immunizations in a specific region and year duration. The results showed that BCG and influ vaccination cause frequent ARs, which is not surprising. The authors also concluded that EPI vaccines in the local region are highly safe, with minimal AR incidences in the general population. The reviewer has some concerns and suggestions for authors to improve their paper.

1) It is not clear why the specific year duration, 2014-2019, was selected in this paper. It is better to expand a little bit to update the data, e.g. 2014-2023.

Response: We chose the 2014-2019 period as it provides a focused examination of adverse events related to vaccines routinely used before the widespread introduction of COVID-19 vaccines. During the pandemic, a substantial number of adverse event claims were specifically related to COVID-19 vaccines, which significantly altered the landscape of vaccine safety data. To maintain clarity and analytical rigor, we have opted to analyze and present the COVID-19 vaccine data in a separate study, ensuring a dedicated and thorough examination of this unique set of vaccines and their associated adverse events. This approach allows us to provide a clear, focused analysis of the traditional vaccines' safety profiles during the study period, without the confounding influence of the unprecedented vaccine rollout and response during the pandemic years. Thank you for your understanding.  

2) The data presentation is not state-of-the-art. The authors should consider updating the style for most of the figures used in this paper.

Response: Thanks for the suggestion. We re-edited the figures in the revised manuscript.

3) Most of the results lack statistical analysis to state the significance of the dataset and results. This is important to convince the reviewers and readers. 

Response: Thank you for your observation regarding the statistical analysis in our study. We understand the importance of robust statistical evaluation to substantiate our findings.

Our study primarily utilizes descriptive statistics to provide a clear overview of the incidence and types of adverse reactions associated with vaccines in Taiwan. The focus on descriptive statistics allows us to effectively communicate the frequency and nature of these events without complex inferential statistics, as our objective is to present an observational account of recorded adverse events.

However, we recognize the value of incorporating additional statistical analyses to enhance the interpretation of our dataset. We are exploring ways to integrate inferential statistics, where appropriate, to better assess the significance of our findings and provide stronger support for our conclusions. This enhancement will help clarify any observed patterns and improve the overall credibility of our results for reviewers and readers alike. Thank you for highlighting this crucial aspect, and we are committed to addressing it to strengthen our study.

4) The data itself is also questionable, as also stated by the authors in 'Limitation section'. It is not clear how the data was collected through which kinds of guidelines and requirements which must be clearly explained in the M&M section.

Response: Thank you for your valuable feedback regarding the data collection process. We appreciate your concern and have taken steps to clarify these aspects in our study.

The data used in our analysis was sourced from the Vaccine Injury Compensation Program (VICP) database, which systematically records comprehensive details about each claim, including the claimant's age, sex, residence, vaccine type, date of immunization, reported symptoms, clinical diagnoses, the interval from vaccination to symptom onset, adjudication outcomes, and compensation amounts. However, we acknowledge that the lack of a standardized reporting form in the VICP can result in variability in the description of symptoms and the sites affected (Discussion, lines 375-382). This variability can sometimes impede consistent analysis.

To address this, we have expanded the Methods and Materials (M&M) section (lines 96-113)  in the revised manuscript to clearly explain the guidelines and procedures for data collection used by the VICP. By detailing these procedures, we aim to provide a clearer understanding of how the data was compiled and the inherent limitations involved. We believe this additional context will help reviewers and readers better assess the robustness and reliability of the data used in our study. Thank you for pointing this out, and we are committed to providing a thorough and transparent depiction of our methodology.

5) It seems that certain conclusions are being exaggerated solely on the basis of datasets that have relatively lower quality. The revised version needs to re-review the dataset and draw proper ones.

Response: Thank you for your feedback on the conclusions drawn from our dataset. We appreciate your concern and would like to address the perceived limitations.

While we acknowledge that some inherent limitations are associated with any observational dataset, we believe the dataset from the Vaccine Injury Compensation Program (VICP) is the most comprehensive source of adverse event data available in Taiwan. It encompasses a wide range of demographic and clinical information related to vaccine-related claims, thus providing a valuable resource for understanding vaccine safety within the country.

Our conclusions are based on this extensive dataset, which, despite its limitations, offers significant insights into adverse events post-vaccination across diverse populations and vaccine types. We have taken care to ensure that our interpretations are cautious and grounded in the data presented. However, we are open to refining our conclusions to more accurately reflect the dataset's strengths and limitations (lines 395-415).

We regard the findings as valuable contributions to the literature on vaccine safety, offering important context and data for public health policy and vaccine monitoring programs in Taiwan and potentially informing practices in similar settings globally. We believe these insights merit publication but are prepared to conduct further reviews and incorporate any necessary adjustments to enhance the clarity and accuracy of our conclusions. Thank you for highlighting these important considerations.

6) Language and writing style need significant improvements. 

Response: The language edition is done by the AI tool ‘Grammarly version 1.87.1.0’.

Round 2

Reviewer 1 Report

Comments and Suggestions for Authors

The revision of the manuscript including suggested corrections has been performed by the authors and the current version seems to be satisfactory.

Author Response

Comments: The revision of the manuscript including suggested corrections has been performed by the authors and the current version seems to be satisfactory.

Response: Thank you for your valuable feedback to enhance the manuscript. We have made further efforts to improve the quality of all figures. The figures are now more clearly visualized.

Reviewer 2 Report

Comments and Suggestions for Authors

The authors have provided explanations for most of the comments raised in this study, and while some of the suggested modifications have been incorporated, they have not significantly improved the overall quality of the manuscript.

Minor Comment: The quality of the charts and figures presented in this study is still poorly represented. These charts need to be improved and presented more effectively.

Comments on the Quality of English Language

Minor editing of English language required.

Author Response

Comments: 

The authors have provided explanations for most of the comments raised in this study, and while some of the suggested modifications have been incorporated, they have not significantly improved the overall quality of the manuscript.

Minor Comment: The quality of the charts and figures presented in this study is still poorly represented. These charts need to be improved and presented more effectively.

Thank you for your valuable feedback on enhancing the manuscript. We have made additional efforts to improve the presentation of tables and figures. The figures are now more clearly visualized, and the tables should be much easier to read.

Reviewer 4 Report

Comments and Suggestions for Authors

This reviewer has no further comments on the revised version of the manuscript. 

Comments on the Quality of English Language

AI tools might be not sufficient sometimes. Corrections might be needed during the proofreading stage.

Author Response

This reviewer has no further comments on the revised version of the manuscript. 

Thank you for your valuable feedback on enhancing the manuscript. We have made additional efforts to improve the presentation of tables and figures. The figures are now more clearly visualized, and the tables should be much easier to read.

Round 3

Reviewer 2 Report

Comments and Suggestions for Authors

The authors have made a concerted effort to address the majority of the comments raised in this study, and the suggested modifications have been incorporated to enhance the quality of the article. The revised manuscript is now clear, concise, and well-written. Additionally, the authors have provided clearer and more readable images for the recommended figures in this updated version.

Comments on the Quality of English Language

Minor editing of English language required.